# A Delphi consensus on the management of oral anticoagulation in patients with non-valvular atrial fibrillation in Spain: ACOPREFERENCE study

Carlos Escobar[1]*, Xavier Borrás[2], Ramón Bover Freire[3], Carlos González-Juanatey[4], Miren Morillas[5], Alfonso Valle Muñoz[6], Juan José Gómez-Doblas[7]

1 Cardiology Department, Hospital Universitario La Paz, Madrid, Spain, 2 Cardiology Department, Hospital de la Santa Creu i Sant Pau, Barcelona, Spain, 3 Cardiology Department, Hospital Clínico San Carlos, CIBERCV, Madrid, Spain, 4 Cardiology Department, Hospital Universitario Lucus Augusti, Lugo, Spain, 5 Cardiology Department, Hospital de Galdakao, Galdakao, Bizkaia, Spain, 6 Cardiology Department, Hospital Marina Salud, Denia, Alicante, Spain, 7 Cardiology Department, Hospital Universitario Virgen de la Victoria, CIBERCV, Málaga, Spain

* escobar_cervantes_carlos@hotmail.com

**Data Availability Statement:** All relevant data are within the manuscript and its Supporting Information files.

## Abstract

### Objective

To evaluate the level of agreement between cardiologists regarding the management of oral anticoagulation (OAC) in patients with non-valvular atrial fibrillation (NVAF) in Spain.

### Materials and methods

A two-round Delphi study was performed using an online survey. In round 1, panel members rated their level of agreement with the questionnaire items on a 9-point Likert scale. Item selection was based on acceptance by $\geq$66.6% of panellists and the agreement of the scientific committee. In round 2, the same panellists evaluated those items that did not meet consensus in round 1.

### Results

A total of 238 experts participated in round 1; of these, 217 completed the round 2 survey. In round 1, 111 items from 4 dimensions (Thromboembolic and bleeding risk evaluation for treatment decision-making: 18 items; Choice of OAC: 39 items; OAC in specific cardiology situations: 12 items; Patient participation and education: 42 items) were evaluated. Consensus was reached for 92 items (83%). Over 80% of the experts agreed with the use of DOACs as the initial anticoagulant treatment when OAC is indicated. Panellists recommended the use of DOACs in patients at high risk of thromboembolic complications (CHA$_2$DS$_2$-VASc $\geq$3) (83%), haemorrhages (HAS-BLED $\geq$3) (89%) and poor quality of anticoagulation control (SAMe-TT$_2$R$_2$ >2) (76%), patients who fail to achieve an optimal therapeutic range after 3 months on VKA treatment (93%), and those who are to undergo cardioversion (80%). Panellists agreed that the efficacy and safety profile of each DOAC

**Funding:** This study was funded by Boehringer Ingelheim. Boehringer Ingelheim participated in the study design and decision to publish, but this company had no role in data collection and analysis or in preparation of the manuscript. No additional external funding received for this study.

**Competing interests:** CE has served on advisory boards for Boehringer, Pfizer/Bristol-Myers Squibb (BMS), Bayer and Daiichi Sankyo, and also received speaker fees from these companies. RBF has served on advisory boards for Boehringer, Novartis and Astra-Zeneca. CG-J has served on advisory boards for Boehringer, Astra-Zeneca, Novartis, and Bayer, and has also received research funding from Abbott, Boehringer, Astra-Zeneca, and Pfizer. AVM has served on advisory boards for MSD, Boehringher, Bayer, Daiichi, and Novartis; and has received honoraria for providing expert testimony from MSD, Boehringher, Bayer, Daiichi, Novartis, Amgen, Sanofi, Pzifer, Janssen, Astrazeneca, Servier, and Rovi. JJGD has received honoraria for advisory services from Bayer, Astrazeneca, MSD, Daiichi, BMS, Amgen and Sanofi. The other authors declare that they have no other competing interests. There are no patents, products in development or marketed products to declare. This does not alter our adherence to Plos One policies on sharing data and materials.

(98%), the availability of a specific reversal agent (72%) and patient's preference (85%) should be considered when prescribing a DOAC. A total of 97 items were ultimately accepted after round 2.

## Conclusions

This Delphi panel study provides expert-based recommendations that may offer guidance on clinical decision-making for the management of OAC in NVAF. The importance of patient education and involvement has been highlighted.

## Introduction

Atrial fibrillation (AF) is the most common sustained cardiac arrhythmia, occurring in approximately 2% of the general population [1]. Its prevalence is strongly associated with age [2], affecting 4.4% of adults over 40 years of age and 17.7% of patients aged 80 or older in Spain [3].

AF is a leading cause of increased morbidity and mortality from ischemic stroke and systemic thromboembolism [4]. AF is associated with a fivefold increase in the risk of thromboembolic stroke [5]. Decreasing the risk of stroke is therefore essential in the clinical management of AF patients. Anticoagulant therapy represents the mainstay for the prevention of stroke and systemic embolism in patients with AF [6, 7].

Vitamin K antagonists (VKAs) have been used for decades as the cornerstone of stroke prevention in non-valvular atrial fibrillation (NVAF). VKAs have widely demonstrated efficacy in reducing stroke or systemic embolism and mortality in AF [8]. However, treatment with VKAs is associated with several limitations such as their narrow therapeutic range which requires frequent monitoring of coagulation parameters, numerous food and drug interactions, and a significant risk of bleeding, including intracranial haemorrhage (ICH) [9].

Direct-acting oral anticoagulants (DOACs) that directly inhibit the activity of thrombin, such as dabigatran, or factor Xa, such as rivaroxaban, apixaban and edoxaban [10, 11] have emerged as therapeutic alternatives for stroke prevention in NVAF. These agents overcome many of the inherent disadvantages of VKAs. Thus, in contrast to VKAs, DOACs have a predictable pharmacodynamic effect, which eliminates the need for routine international normalised ratio (INR) testing [12]. DOACs have been found to be non-inferior to VKAs in stroke prevention without increasing the risk of major bleeding [13–17]. On the basis of the efficacy, safety and convenient administration of DOACs, the current international guidelines recommend these agents as preferable to VKAs for most patients with NVAF for whom oral anticoagulation (OAC) is indicated [18]. However, the use of VKAs remains significantly more predominant than DOACs in Spain [19] despite the fact that approximately 40% of AF patients on VKA treatment have poor control of anticoagulation [20–22], placing them at higher risk of both embolic and bleeding complications [23]. This situation emphasises the importance of improving the management of anticoagulant therapy for stroke prevention in patients with NVAF.

There are several guidelines available to provide guidance on the management of anticoagulation in AF patients, providing clinicians with evidence-based recommendations for stroke prevention. However, treatment decision-making is often challenging in routine clinical practice, given that guideline recommendations are based on clinical trials where some specific patient profiles are not represented. The role of physicians, based on their daily clinical

practice and knowledge, is essential to fill the gap left by the evidence-based guidelines. When guidelines fail to provide clear direction in certain clinical situations, consensus methods based on expert opinion may provide support to physicians in treatment decision-making. The Delphi technique is a reliable consensus method of gathering expert opinion, which involves an anonymous iterative process comprising a series of feedback rounds until consensus is achieved among a geographically dispersed group of experts [24].

On the basis of this background, we conducted a Delphi method study to address multiple key questions and controversies related to the selection and management of anticoagulation in patients with NVAF. A two-round Delphi survey was used to seek expert-based opinion to develop a set of consensus guidelines that may hopefully provide support to clinicians in clinical decision-making in order to improve real-world practice in these patients.

## Patients and methods

### Study design

The ACOPREFERENCE project was a nationwide Spanish multicentre 2-round Delphi study to seek expert opinion on the management of anticoagulation therapy for patients with NVAF.

The approval of the Institutional Review Board (IRB) or by equivalent ethics committee(s) was not required as this Delphi study does not involve human subjects research. No patient data were collected for this study, which was completely based on the feedback provided by experts regarding relevant topics on anticoagulation management in NVAF.

The Delphi process is a widely accepted scientific method of structured and systematic information gathering from a group of experts (termed the Delphi expert panel) on controversial or complex topics [25]. Each panel expert provides opinions individually and anonymously without the biasing effect of dominant individuals or group pressure [24, 26, 27]. The Delphi process ends when an agreement has been reached on the discussed topics.

A modified Delphi method was used in this study according to the RAND/UCLA recommendations [28]. The Delphi project was carried out in six steps: 1) Literature review (by the scientific committee); 2) Discussion and questionnaire domain/item generation by the scientific committee in a face-to-face meeting; 3) Selection of the Delphi expert panel and invitation to candidates to participate in Delphi process; 4) Domain/item set evaluation by the panel experts through 2 rounds using an online platform through a web platform (two-round Delphi approach); 5) Final discussion of items that did not reach consensus in preceding rounds among the scientific committee experts; 6) Final consensus analysis.

### Delphi process

**Selection of Delphi participants.**   The expert scientific committee was comprised of seven cardiologists experienced in the management of patients with AF and recognised experts in the field.

A total of 250 cardiologists from hospitals distributed across Spain were initially contacted and invited to participate in the project as members of the Delphi expert panel in both round 1 and round 2 of the Delphi process. The panel experts were selected based on their extensive experience in and knowledge of the management of patients with NVAF and anticoagulation.

The expert panel members were provided with an informative leaflet outlining the aims and the study procedure and including an electronic link to the online survey. Experts received personalised access to the online survey. Panel members participated in the project through 2 rounds of the Delphi process using the online questionnaire. The purpose of the expert panel was to reach a consensus based on the current clinical evidence and their daily practice in and knowledge of the management of anticoagulation therapy in AF.

**Selection of Delphi questionnaire dimensions/items.** The scientific committee carried out a systematic literature review including relevant studies, clinical practice guidelines, and reviews regarding the choice and management of anticoagulant treatment in NVAF focusing on current controversial topics, and patient education and involvement in AF management. After a careful and critical review of the selected literature, and based on their knowledge of the clinical management of anticoagulation in AF, the scientific committee developed the first set of domains and items for the Delphi questionnaire in a face-to-face meeting.

**Round 1.** The members of the Delphi expert panel evaluated the inclusion of the items in the consensus using an online questionnaire through a web platform. Panel members were asked to rate their level of agreement with each questionnaire item on a 9-point Likert scale from 1 (completely disagree) to 9 (completely agree). Panellists were also encouraged to provide comments after scoring each item using open-text comment fields included in the online survey.

After the analysis of the data obtained from the first Delphi round, the scientific committee experts participated in a teleconference meeting, where the Delphi survey results were presented and discussed. Item selection was based on the acceptance of questionnaire items by $\geq$66.6% of the expert panel and the agreement of the scientific committee. Statements not achieving 66.6% agreement were removed or modified according to the feedback provided by the expert panel. All statements were assessed in view of the experts' suggestions. After round 1 was completed and the expert comments had been summarised, amendments were made to some questionnaire items. Redundant statements were grouped and reduced and issues regarding comprehension of some statements were fixed on the basis of the expert comments. Where necessary, new items were generated and included. The updated questionnaire was redistributed to the panellists for round 2.

**Round 2.** In round 2, the same panel members were asked to evaluate the list of items that did not meet consensus from round 1, using the same voting method as described for the preceding round. For this evaluation, the panel members were provided with a summary of the opinions issued anonymously by the participants in the first round, in addition to any other information that the scientific committee deemed appropriate to make available to the panellists in order to achieve consensus. Thus, the panellists could reflect upon the group's responses after the first round and re-evaluate the non-consensus items in view of the other experts' feed-back.

After analysis of the responses as described for round 1, the statements not meeting expert agreement were retained for discussion in round 3.

**Round 3.** Round 3 comprised a teleconference meeting among the scientific committee experts to assess those items that did not reach consensus in round 2. The members of the scientific committee discussed the non-consensus items until agreement was reached to retain or eliminate the item from the final consensus guidelines.

## Statistical analysis

Descriptive statistical analysis of the data obtained from the assessment of the Delphi questionnaire items by the expert panel in rounds 1 and 2 was conducted. The distribution of frequencies of panel responses on the 9-point scale was calculated to establish the level of consensus for each questionnaire item. Each item was categorised according to the scores as rejected (scores 1–3), undetermined (scores 4–6), or accepted (scores 7–9).

A descriptive statistical analysis of the characteristics of the Delphi expert panel was also performed, including calculation of measures of central tendency and dispersion (mean ± standard deviation, median and interquartile range) for quantitative variables, and frequencies and valid percentages for qualitative variables.

The statistical analysis was performed using the Statistical Package for the Social Sciences (SPSS) version 18.0 (SPSS Inc., Chicago, IL, USA).

## Results

### Panel experts

A total of 238 cardiologists from 172 hospitals distributed throughout Spain agreed to participate in the project as Delphi panel experts. All 238 experts participated in round 1. Of these, 217 experts completed the round 2 survey. The characteristics of the Delphi expert panellists are summarised in Table 1. Briefly, most participant experts were associate physicians (88.2%) in public hospitals (87.4%) with a median of 15 years of professional experience. Approximately 60% of the cardiologists were involved in research.

### Results from round 1 and round 2

In round 1, panel members evaluated 111 items from the following 4 dimensions: 1) Evaluation of thromboembolic and bleeding risk for treatment decision-making: 18 items; 2) Choice of anticoagulant treatment for patients with NVAF: 39 items; 3) Patient participation and education: 42 items; 4) Use of anticoagulants in specific cardiology situations: 12 items (S1 Appendix). Consensus was reached for 92 items (83%), which were accepted without modification. After the scientific committee meeting, 9 items were removed and 5 items (4 non-consensus items and one item which had reached consensus from the dimension related to the evaluation of thromboembolic and bleeding risk for treatment decision-making) were grouped and reduced to 2 items to be evaluated in round 2. Thus, when panellists were asked to rate their level of agreement with 3 items regarding the use of ATRIA, HEMORR$_2$HAGES, and ORBIT for haemorrhagic risk assessment, respectively, these items failed to reach consensus given that these scales are generally unknown and underused in routine clinical practice according to the expert panel feed-back. The members of the scientific committee therefore decided to combine

**Table 1. Characteristics of the Delphi expert panel.**

| Characteristics | Value |
|---|---|
| **Age**, median (range), years | 40 (35–49) |
| **Gender**, male, n (%) | 144 (39.5) |
| **Professional experience**, median (range), years | 15 (8.8–21.3) |
| **Hospital position**, n (%) | |
| Service head | 10 (4.2) |
| Department head | 9 (3.8) |
| Associate physician | 210 (88.2) |
| Other [1] | 9 (3.8) |
| **Research experience**, n (%) | 154 (64.7) |
| **Professor**, n (%) | 36 (15.1) |
| **Type of hospital**, n (%) | |
| Public hospital | 208 (87.4) |
| Private hospital | 26 (10.9) |
| Other [2] | 4 (1.7) |
| **University hospital**, n (%) | 119 (50.0) |

[1] Resident: n = 6

[2] Public and private: n = 2; concerted hospitals: n = 2

these items into a single statement. In addition, the initial proposal of dimensions/items contained 2 items wherein experts were asked about the factors which have the greatest weight in the $CHA_2DS_2$-VASc scale, considering age and stroke in 2 different items. The item regarding the weight of stroke in this scale achieved consensus while the item asking about the weight given to age failed to reach consensus. Based on the expert comments regarding the equal weight of both factors for risk assessment, these items were grouped into one single item.

A total of 7 items that did not meet consensus or were grouped or reformulated after round 1 were put forward for inclusion in round 2 along with accompanying comments. Four items reached consensus in round 2. Three items that did not achieve consensus in round 2 were discussed in round 3 by the scientific committee experts with one item being finally accepted. At the end of the Delphi process, a total of 97 items were finally retained. Fig 1 illustrates the results of the modified Delhi study.

Figs 2–5 summarise the results from the Delphi process and the level of agreement after the 2 rounds for the statements related to the evaluation of thromboembolic and bleeding risk for treatment decision-making (Fig 2), the choice of anticoagulant treatment for patients with NVAF (Fig 3), patient participation and education (Fig 4), and the use of anticoagulants in specific cardiology situations (Fig 5).

OAC: oral anticoagulation

[1] The removal of this item was due to the generalised opinion of the panellists of the disuse and lesser discrimination of this scale and the position in favour of the use of the $CHA_2DS_2$-VASc scale.

[2] These items were grouped after round 1. The initial proposal of dimensions/items contained 2 items wherein experts were asked about the factors which have the greatest weight in the $CHA_2DS_2$-VASc scale, considering age and stroke in 2 different items ("Of the criteria considered in the $CHA_2DS_2$-VASc scale, age is the thromboembolic risk factor with the greatest weight"; "Of the criteria considered in the $CHA_2DS_2$-VASc scale, stroke is the thromboembolic risk factor with the greatest weight"). Based on the expert comments regarding the equal weight of both factors for risk assessment, these items were grouped into one single item (item 6) to be evaluated in round 2. This item finally achieved consensus in round 2 after being grouped/reformulated.

[3] In round 1, when panellists were asked to rate their level of agreement with 3 items regarding the use of ATRIA, $HEMORR_2HAGES$ and ORBIT, respectively, these items failed to reach consensus given that these scales are generally unknown and underused for haemorrhagic risk assessment in routine clinical practice according to the expert panel feed-back. Therefore, the members of the scientific committed decided to combine these items into a single statement (item 12) to be evaluated in round 2. This item finally achieved consensus in round 2 after being grouped/reformulated.

[4] This item was deleted given that this scale is unknown and not generally used in routine clinical practice based on the comments made by the expert panel.

[5] This item failed to reach consensus in round 2 and it was finally eliminated after discussion in round 3.

DOAC: Direct-acting oral anticoagulant; ICH: Intracranial haemorrhage; INR: International normalised ratio; NVAF: Non-valvular atrial fibrillation; OAC: Oral anticoagulation; TPR: Therapeutic Positioning Report; VKA: Vitamin K antagonist

[1] This item did not achieve consensus in round 1 and it was finally eliminated.

[2] In spite of good treatment adherence or if VKAs are contraindicated.

[3] Except in the case of ischemic stroke with clinical and neuroimaging criteria of high risk of ICH, severe arterial thromboembolic episodes, or inability to access INR controls.

**Round 1**
**(February 15 - June 3, 2017)**
**238 panelists**

- **111 items** distributed in **4 dimensions**.
- **92 items** achieved **consensus** (83%) (accepted without modification)
  - Evaluation of thromboembolic and bleeding risk: 10/18 items.
  - Choice of anticoagulant treatment for patients with NVAF: 35/39 items.
  - Patient participation and education: 37/42 items.
  - Use of anticoagulants in specific cardiology situations: 10/12 items.
- 9 items removed.
- 5 non-consensus items to be evaluated in round 2.
- 5 items grouped into 2 items to be evaluated in round 2.

**Round 2**
**(June 27 -September 25, 2017)**
**217 panelists**

- **7 items** wich did not achieve consensus or were modified in round 1.
- **4 items** reached **consensus**
  - Evaluation of thromboembolic and bleeding risk: 2/3 items.
  - Choice of anticoagulant treatment for patients with NVAF: 1/1 item.
  - Patient participation and education: 1/3 items.
- 3 items to be evaluated in round 3.

**Round 3**
**(October 10 - December 15, 2017)**
**Scientific committee**

- Discussion of **3 items** which did not achieve consensus in round 2.
- **1 items accepted**/2 items removed.

**Final guideline document**

97 consensus items

**Fig 1. Results of the modified Delphy study.**

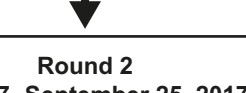

(4) Item 10: Known hypersensitivity or specific contraindication to the use of acenocoumarol or warfarin; Item 11: History of ICH if it is assessed that the benefits of anticoagulation outweigh the risk of bleeding; Item 12: Patients with ischaemic stroke who meet clinical and neuroimaging criteria for high risk of ICH, defined as the combination of HAS-BLED ≥3 and grade III-IV leukoaraiosis and/or multiple cortical microbleeds; Item 13: Patients on treatment with VKAs who suffer severe arterial thromboembolic episodes despite having good INR control; Item 14: Patients who have started treatment with VKAs in whom INR control cannot be

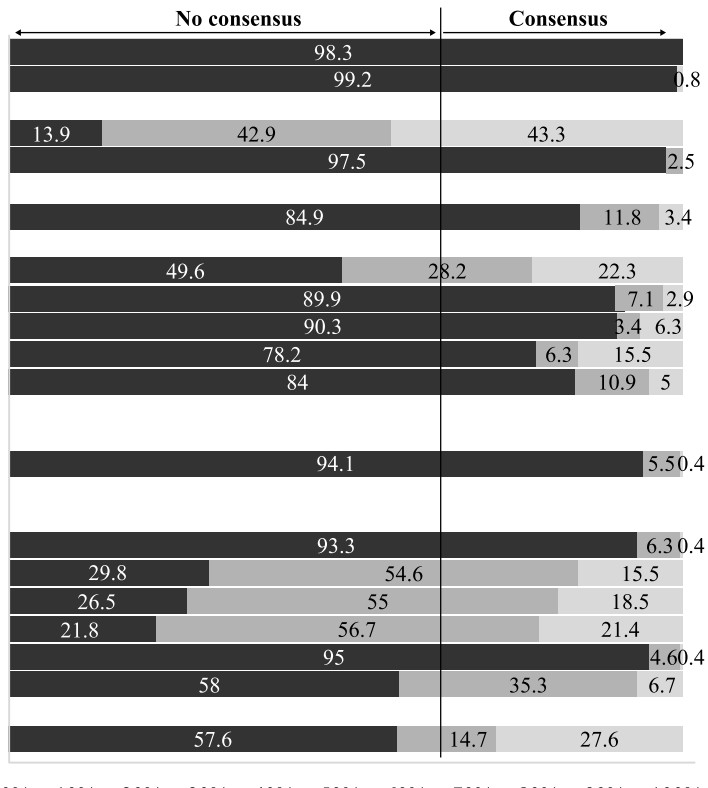

**Round 1**

1. Reduction of the risk of stroke is the primary reason for initiating OAC
2. Individualization of treatment decision should be based on thromboembolic and bleeding risk
3. $CHADS_2$ is the most appropriate scale to evaluate thromboembolic risk[1]
4. $CHADS_2DS_2$-VASc is the most appropriate scale to evaluate thromboembolic risk
5. $CHA_2DS_2$-VASc has a greater discriminative capacity to identify low thromboembolic risk
6. Age has the greatest weight in the $CHA_2DS_2$-VASc scale[2]
7. Stroke has the greatest weight in the $CHA_2DS_2$-VASc scale[2]
8. Anticoagulation is recommended in male patients with $CHA_2DS_2$-VASc ≥2
9. Anticoagulation is recommended in female patients with $CHA_2DS_2$-VASc ≥3
10. Anticoagulation should be considered in females and males with a $CHA_2DS_2$-VASc score of 2 and 1 respectively after assessing the risk-benefit ratio and the patient's preference
11. Bleeding risk assessment scales should be considered to identify modifiable risk factors
Bleeding risk should be assessed using (items 12-15):
12. HAS-BLED scale
13. ATRIA scale[3]
14. $HEMORR_2HAGES$ scale[3]
15. ORBIT scale[3]
16. A HAS-BLED score ≥3 indicates a high bleeding risk
17. The $SAMe-TT_2R_2$ scale should be used to predict poor INR control when deciding treatment[4]
18. The $SAMe-TT_2R_2$ scale is routinely used for patient management[5]

**Round 2**

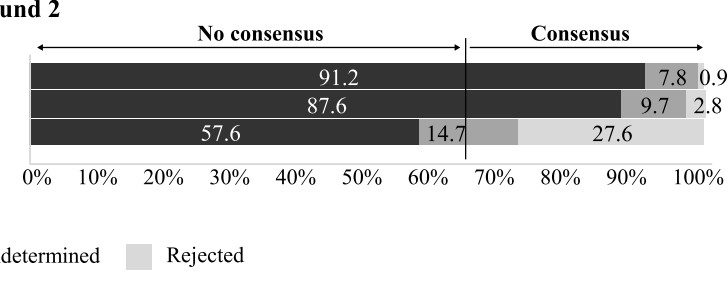

6. Age and stroke have the greatest weight in the $CHA_2DS_2$-VASc scale[2]
12. The ATRIA, ORBIT and $HEMORR_2HAGES$ are little used[3]
15. The $SAMe-TT_2R_2$ scale is routinely used for patient management[5]

■ Accepted    ▨ Undetermined    ▢ Rejected

**Fig 2. Results of the two-step Delphi process for the items relating to the evaluation of thromboembolic and bleeding risk for treatment decision.**

maintained within the therapeutic range (2–3) despite good compliance, considering that INR control is inadequate when TTR calculated by the Rosendaal method is <65% when this method is available or the percentage of INR values within the therapeutic range (direct TRT) is <60%, considering an assessment period of 6 months in both cases, excluding INR values of the first month or periods of change resulting in modification of the VKA regimen.

[5] Clinical situations detailed in items 10–14.

[6] This item achieved consensus in round 2 (Accepted: 71.9%; Undetermined: 20,7%; Rejected: 7,4%).

DOAC: Direct-acting oral anticoagulant; OAC: Oral anticoagulation

[1] This item was modified after round 1 to be evaluated in round 2. The item finally achieved consensus in round 2 after being modified.

[2] This item did not achieve consensus in round 1 and it was finally eliminated.

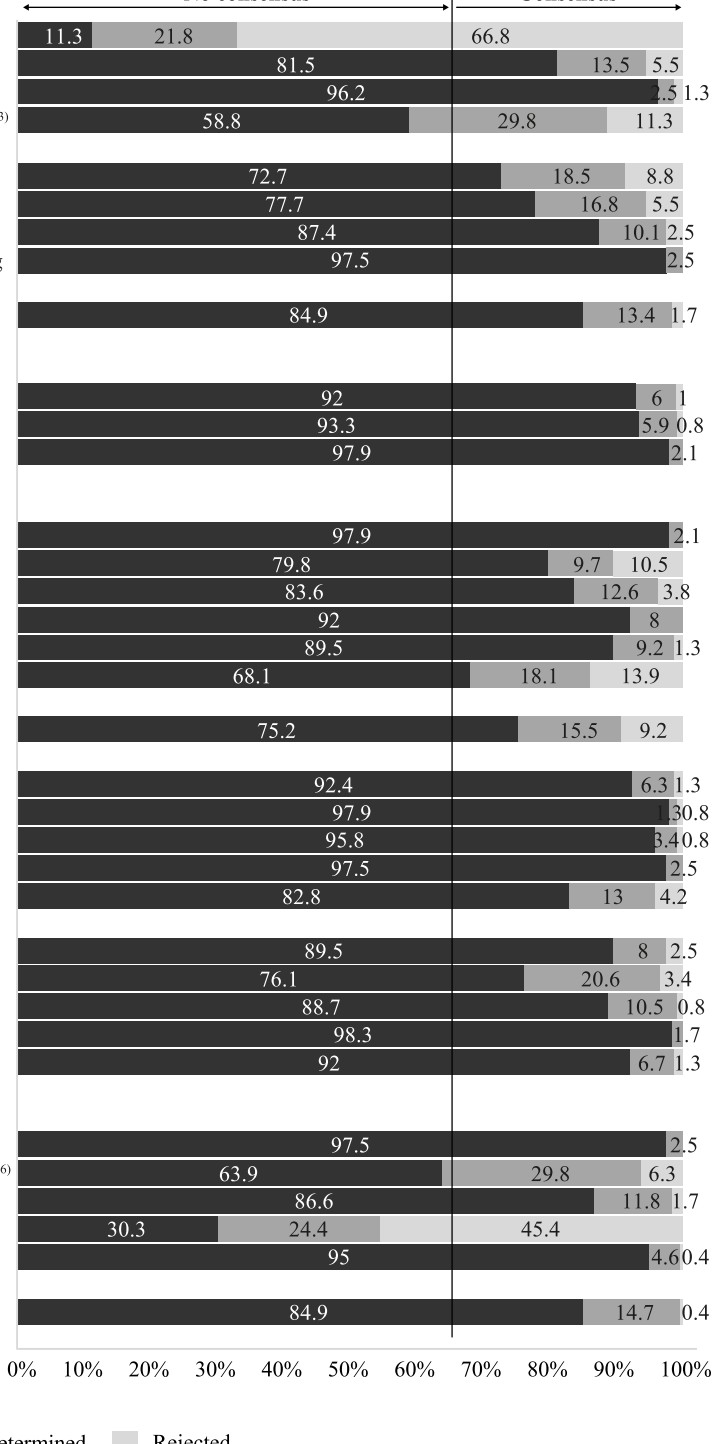

**Round 1**

1. VKAs should be the first-choice therapy when anticoagulation is indicated[1]
2. DOACs should be the first-choice therapy when anticoagulation is indicated
3. DOACs should be considered when INR control is suboptimal with VKAs[2]
4. Switching fromVKAs to a DOAC in controlled patients is not recommended[3]
DOACs are a cost-effective alternative to classic OAC in (items 5-7):
5. All patients with NVAF
6. Specific profiles of patients with NVAF
7. Patients at high thromboembolic risk
8. DOACs are recommended for patients with high thromboembolic or bleeding risk or poor INR control
9. The use of VKAs remains significantly more predominant than DOACs in Spain
The use of DOACs would be a suitable option in the following situations:
10-14. In the situations stated in the TPR (detailed in items10-14)[4]
15. Failure to achieve an optimal therapeutic range within 3 months
16. Access to conventional INR control is not possible
To initiate treatment with DOACs, patients should meet the following criteria (items17-21):
17. No specific contraindications for DOACs
18. At least one of the following previously mentioned clinical situations[5]
19. History of previous good treatment compliance
20. Ability of the patient and/or family to understand the risk-benefit ratio
21. Reliable possibility of periodic follow-up (renal function monitoring, etc)
22. All the criteria stated in the TPR must be met before DOACs can be prescribed in a patient who is candidate for OAC
23. There are unidentified situations in the TPR where the benefit of DOACs could be superior toVKAs
24. The risk of severe bleeding is lower with DOACs than with VKAs
25. The bleeding with greatest risk is ICH
26. The risk of ICH is lower with DOACs than with VKAs
27. It is reasonable that DOACs are used in patients at a greater risk of ICH
28. Males and females with a $CHA_2DS_2$-VASc score ≥3 and ≥4 respectively should be candidates for DOACs
29. Patients with high bleeding risk should be candidates for DOACs
30. Patients with a $SAMe-TT_2R_2$ ≥2 should be candidates for DOACs
31. DOACs are a suitable alternative for polymedicated patients
32. Assessment of renal function is essential when prescribing DOACs
33. During treatment with DOACs, renal function should be assessed at least once a year
When prescribing a DOAC (items 34-36)
34. The efficacy and safety profile of each DOAC should be considered
35. It should be taken into account whether a specific reversal agent is available[6]
36. It is essential to assess the expected treatment adherence
37. Patients treated with DOACs could have lower treatment adherence[1]
38. Proper education should be ensured by an adequate explanation of the disease and the benefits and risks of OAC
39. After explaining the different options for OAC, the patient's preference should be taken in to account

**Fig 3. Results of the two-step Delphi process for the items relating to the choice of anticoagulant treatment in patients with NVAF.**

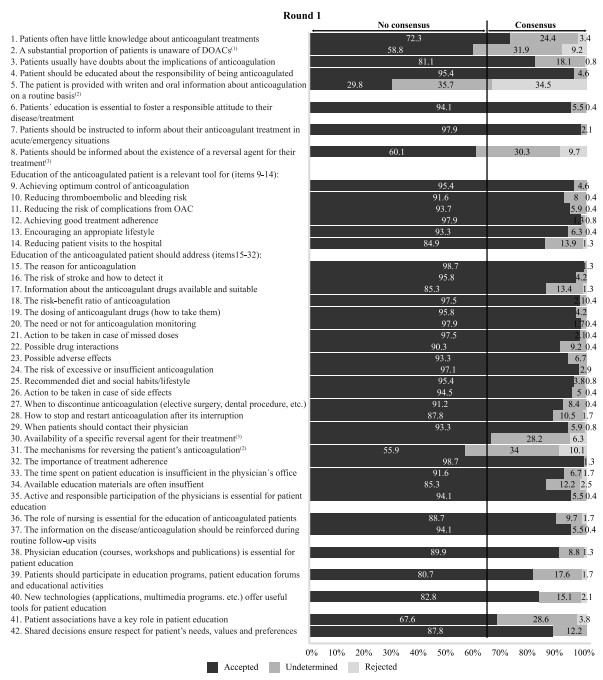

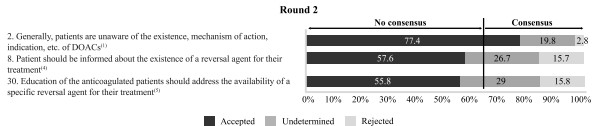

**Fig 4. Results of the two-step Delphi process for the items relating to the participation and education of the anticoagulated patient.**

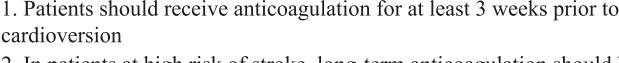

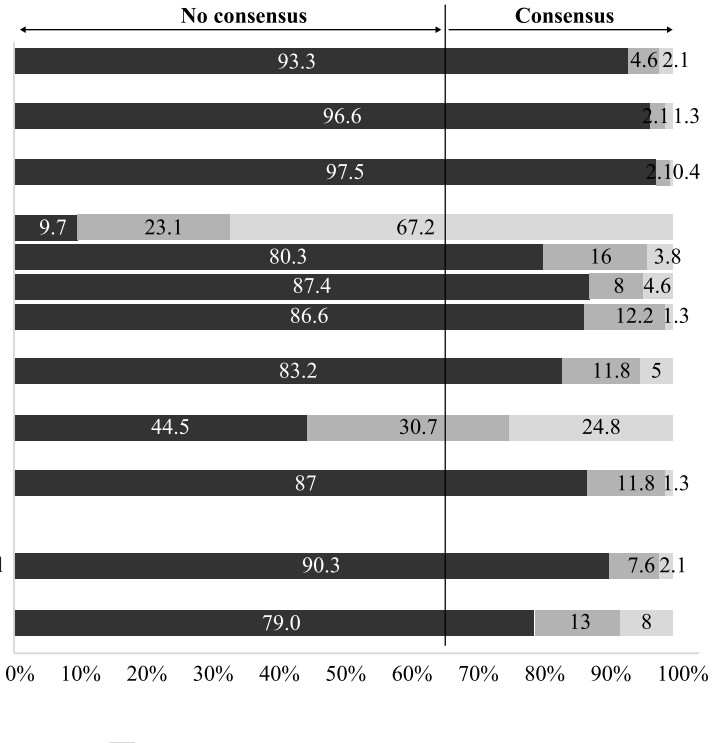

**Fig 5. Results of the two-step Delphi process for the items relating to the use of anticoagulants in specific cardiology situations.**

[3] This item did not achieve consensus in round 1 and it was evaluated in round 2.

[4] This item failed to achieve consensus in round 2, but it was finally accepted after discussion in round 3.

[5] This item failed to reach consensus in round 2, and it was finally eliminated after discussion in round 3.

ACS: Acute coronary syndrome; DOAC: Direct-acting oral anticoagulant; OAC: Oral anticoagulation; SCD: Stable coronary disease; VKA: Vitamin K antagonist

[1] According to the recommendations for long-term anticoagulation, regardless of the method of cardioversion or maintenance of sinus rhythm.

[2] This item did not achieve consensus in round 1 and it was finally eliminated.

With the aim of simplifying the interpretation of the results, a set of recommendations was developed with the most relevant items that reached expert consensus in the Delphi process (Table 2).

**Table 2. Expert-based Delphi consensus recommendations for the management of oral anticoagulation in patients with NVAF.**

1. The $CHA_2DS_2$-VASc score is the most appropriate method to assess thromboembolic risk in patients with NVAF who are candidates for oral anticoagulant treatment.

2. Initiation of anticoagulant therapy should be considered in patients with a $CHA_2DS_2$-VASc score of 2 in female patients and 1 in male patients after assessing the risk-benefit ratio and the patient's preference.

3. Bleeding risk should be assessed in patients with NVAF using the HAS-BLED scale, with a patient at high bleeding risk being considered as one having a HAS-BLED score $\geq$3.

4. Treatment with DOACs should be the initial anticoagulant therapy in those patients with NVAF in whom anticoagulation is indicated.

5. The use of DOACs would be a suitable option in patients with NVAF who are candidates for anticoagulant therapy in the following situations:

 a. Patients at high risk of thromboembolic complications and haemorrhages.

 b. Male patients with $CHA_2DS_2$-VASc $\geq$3 and female patients with $CHA_2DS_2$-VASC $\geq$4.

 c. Patients who have failed to achieve an optimal therapeutic range for INR within 3 months from the start of VKA treatment.

 d. Patients with a score greater than 2 on the SAMe-$TT_2R_2$ scale (high risk of poor quality of anticoagulation control with VKAs)

 e. Patients who are to undergo cardioversion. The lack of control of VKAs makes their use difficult prior to cardioversion.

6. The efficacy and safety profile of each DOAC should be taken into account when prescribing a DOAC.

7. After explaining to the patient the different options for oral anticoagulation (VKAs and DOACs), the patient's preference should be taken into account when prescribing anticoagulant treatment for stroke prevention.

8. Patient education is essential to foster in the anticoagulated patient a responsible attitude to their disease and their treatment. In addition, patient education is a relevant tool for achieving optimum control of anticoagulation, reducing thromboembolic and bleeding risk, achieving good treatment adherence, encouraging a lifestyle appropriate to their condition, and reducing patient visits to the hospital.

9. Patients education should address: the reason for anticoagulation, the risk of stroke and how to detect it, information about the anticoagulant drugs available and suitable for their case, the risk-benefit ratio of anticoagulation, the dosing of anticoagulant drugs (adequate explanation about how to take them), the need or not for anticoagulation monitoring, action to be taken in case of missed doses, possible drug interactions with the treatment, recommended diet and social habits/lifestyle, when and how to stop and restart anticoagulant treatment in specific situations in which anticoagulant treatment should be interrupted (elective surgery, dental procedure, etc.), and the importance of treatment adherence.

10. Patients, supported by healthcare professionals (physicians and nursing staff), should participate in the management of their disease through health education programs, patient education forums and educational activities (courses, workshops, conferences, etc.).

11. Active and responsible participation of physicians and physician education through attendance of courses, workshops and publications related to anticoagulant therapy is essential for patient education.

DOAC: Direct-acting oral anticoagulant; INR: International normalised ratio; NVAF: non-valvular atrial fibrillation: TPR: Therapeutic Positioning Report; VKA: Vitamin K antagonist

## Discussion

The present Delphi panel study showed a high level of agreement between national cardiology experts regarding relevant topics on anticoagulation management in patients with NVAF. A high level of consensus was reached regarding thromboembolic and haemorrhagic risk assessment tools to be used for treatment decision-making, selection criteria for anticoagulant treatment, and anticoagulation management in special cardiology situations. The importance of patient education and involvement was emphasised. Expert panel opinions were generally in line with current clinical practice guidelines on anticoagulation in NVAF. However, this study revealed that there are controversial issues regarding OAC management that need to be standardised and some recommendations based on new scientific evidence should be adopted.

There was a clear consensus on the individualisation of anticoagulant treatment based on the thromboembolic and haemorrhagic risk of the patient. The panel unanimously agreed that $CHA_2DS_2$-VASc instead of $CHADS_2$ is the most appropriate scale to evaluate thromboembolic risk of patients with NVAF who are candidates for OAC. The panellists concurred that the $CHA_2DS_2$-VASc scale better predicts thromboembolic events among those patients with a lower risk score [29]. Accordingly, $CHA_2DS_2$-VASc has become the preferred risk assessment tool in clinical decision making [18, 30]. It was agreed that patients with a $CHA_2DS_2$-VASc score of 2 in female patients and 1 in male patients should start anticoagulant therapy after assessing the risk-benefit ratio and the patient's preference as recommended by the 2016 European Society of Cardiology (ESC) guidelines [18].

More than 90% of panellists supported the use of bleeding scoring scales to identify risk factors that should be modified in patients who are candidates for OAC according to the European guidelines [18]. The expert panel agreed that haemorrhagic risk should be assessed with the HAS-BLED scale [31] which has been validated in a wide range of patients and it is the only bleeding risk scale that is predictive of ICH [32], the most serious bleeding complication due to its high risk of mortality or subsequent major disability [33]. Indeed, HAS-BLED score has been shown to have the best predictive performance for major bleeding compared to the $HEMORR_2HAGES$ and ATRIA scoring tools [34] which are rarely used in routine clinical practice according to our findings.

The Delphi panel unanimously agreed that DOACs should be the initial anticoagulant therapy in patients with NVAF in whom OAC is indicated according to the ESC guidelines [18], which strongly recommend DOACs over VKAs. However, although the need for DOACs was emphasised, the panellists acknowledged that the use of VKAs remains significantly more predominant than DOACs in Spain. The prescription of DOACs is still a challenge in Spain due to the gap between evidence-based guidelines and clinical practice, which is currently limited by administrative and bureaucratic barriers to DOAC prescription. The use of DOACs in AF patients is subject to the requirement of prior authorisation by the Spanish National Health System which involves the prescription to be validated before it is accepted for funding and dispensing. This situation results in underuse or delayed prescription of DOACs in patients who could benefit from these agents. In addition, the prescription of DOACs is limited by the restrictive criteria stated in the Therapeutic Positioning Report (TPR) released by the Spanish Agency of Medicines (AEMPS for its acronym in Spanish) [35].

The participating cardiologists concurred that all the criteria established by the TPR must be met for the use of DOACs in patients with NVAF: VKA contraindication, history of ICH, ischemic stroke with clinical and neuroimaging criteria for high risk of ICH, arterial thromboembolic episodes despite having good INR control with VKAs, suboptimal INR control with VKAs despite good compliance, and inability to access INR controls. However, the national TPR does not take into account the updated recommendations of the ESC guidelines [18] and

the relevant new scientific evidence on key topics regarding the use of DOACs in NVAF. Accordingly, the panellists agreed that there are some clinical situations in which DOACs would be a more suitable option than VKAs; however, these situations are not addressed in the current guidelines for OAC use. Thus, there was a clear consensus on the use of DOACs in patients at high risk of thromboembolic complications or haemorrhages as recommended by the consensus document developed by the Spanish Society of Cardiology (SEC for its acronym in Spanish) [36]. Accordingly, the panellists recommended that male patients with $CHA_2DS_2$-VASc $\geq 3$ and female patients with $CHA_2DS_2$-VASC $\geq 4$ should be candidates to receive DOACs. In addition, nearly 90% of panel experts supported that patients at high bleeding risk (HAS-BLED scale >3), should be treated with DOACs. Surprisingly, a high risk of bleeding is not currently considered to be a criterion for receiving treatment with DOACs despite the fact that these agents have demonstrated a superior safety profile, particularly in terms of ICH. Indeed, a superior net clinical benefit of DOACs over VKAs has been demonstrated in patients with a high risk of bleeding [37].

Predicting the quality of anticoagulation in patients on VKAs may prevent the occurrence of thromboembolic and haemorrhagic events. The SAMe-$TT_2R_2$ scale has been proposed as an adequate and reliable method to identify patients most likely to have poor anticoagulation control [38]. This scale has been validated in Spain, showing a good correlation between the score and the TTR in addition to an association with the occurrence of events in patients treated with VKAs [39, 40]. In line with the SEC's recommendations, the expert panel agreed that patients with a SAMe-$TT_2R_2$ score >2 should be candidates to receive initial anticoagulant therapy with DOACs due to their high probability of having suboptimal anticoagulation with conventional anticoagulant drugs. However, as the panellists acknowledged, this scale is not usually used in routine clinical practice. The SAMe-$TT_2R_2$ score should therefore be adopted in real-world practice given that its application would avoid the use of VKAs in those patients more likely to have poor anticoagulation control during the first 6 months, which is clearly associated with higher risk of thromboembolic or haemorrhagic events [23]. In this regard, panel experts also agreed that the quality of anticoagulation measured by the TTR should be achieved within 3 months from the start of treatment with VKAs instead of 6 months, as recommended by current ESC guidelines (18), probably based on an increased risk of thromboembolic and haemorrhagic complications [41] during the first few months after OAC initiation [41, 42].

Anticoagulation is a cornerstone of peri-cardioversion management in AF patients to reduce the risk of thromboembolic events. However, VKAs' lack of control was recognised by the panellists as a limitation for the use of VKAs in patients undergoing cardioversion. According to the Delphi experts, DOACs should be the treatment of choice in patients who are scheduled to undergo cardioversion according to the available scientific evidence which has provided insights in favour of DOACs as an alternative strategy to VKA therapy for peri-cardioverison thromboembolic prophylaxis [43].

The vast majority of experts concurred that the efficacy and safety profile of each DOAC and the expected degree of therapeutic compliance are key factors that should be taken into account when prescribing a DOAC. Panellists also agreed that the assessment of renal function is essential when prescribing DOACs given that each of these agents to a varying degree is eliminated by the kidneys [44], and supported the recommendation of the European guidelines of evaluating renal function at least once a year to adjust doses and redefine the risk [18].

The availability of specific reversal agents seems to remain as a criterion for DOAC prescription according to the results generated by the experts. Indeed, universal adoption of DOACs has been stunted by the lack of specific antidotes. However, based on the comments of the experts, these findings are likely to be due to the need of antidotes to reverse the effect of

DOACs in case of high risk of haemorrhage, or if surgical intervention is required emergently. In addition, with the availability of idarucizumab to reverse the effects of the direct thrombin inhibitor dabigatran [45, 46] and other antidotes being marketed in the near future for factor Xa inhibitors [47], concerns regarding the management of bleeding complications may no longer be warranted in the prescription of DOACs, where appropriate.

The panellists also considered that the patient's preference between VKAs and DOACs should be taken into account at the time of OAC prescription. Patients should be provided with enough information so that they can decide about the best anticoagulation treatment option. However, as the panellists recognised, patients have limited knowledge about the available anticoagulant treatment options, with a high proportion of patients who are unaware of the existence or indication of DOACs. The importance of patient education when initiating DOAC therapy was emphasised. The findings of the present Delphi study recommended that patients, supported by healthcare professionals, should participate in the management of their disease and treatment through health education programs, patient education forums and educational activities. The education of patients is a key issue in the core process to encourage a self-management role and to empower patients to participate in shared decision-making [48]. Most experts supported shared decision-making in line with the European guidelines which recommend the central role of the patient in tailoring management to patient preferences and improving adherence to long-term therapy [18]. However, participant cardiologists encountered difficulties in patient education during the daily management of AF patients given that they do not always have sufficient adequate patient education materials on hand in the office and the time available for patient education is insufficient.

The main limitations of this study arise from the obvious concerns with regard to a Delphi panel study such as the potential bias derived from the selection of experts and the subjectivity linked to the potentially divergent personal opinions of the panellists which may partly result from unevenly distributed expertise. Nevertheless, this study is strengthened by the large number of experts participating through 2 rounds of the Delphi process to reach consensus. Approximately 200 experts involved in the management of AF patients in hospitals distributed homogeneously throughout the country have contributed to the Delphi survey. In addition, a high level of consensus was achieved on most items included in the Delphi questionnaire, which supports the importance of taking into account the consensus recommendations (Table 2) at the time of treatment decision-making and during the routine follow-up visits.

Despite the abovementioned limitations, this modified Delphi study provides useful expert-based consensus recommendations that may improve the management of patients with NVAF in routine clinical practice. To our knowledge, this is the first Delphi method study aiming to achieve consensus among cardiologists on the selection and management of OAC for patients with NVAF in Spain.

The consensus guidelines emerging from the large panel of experts in the ACOPREFERENCE Delphi study may help to standardise the use of OAC for patients with NVAF and support clinicians in treatment decision-making. Therefore, the clinical practice guidelines obtained may hopefully improve patient care in these patients, including challenging cases which have been omitted from current national guidelines for OAC use. In addition to providing guidance on clinical decision-making, this study has highlighted the importance of patient education and provided recommendations to improve patient involvement and self-management.

## Supporting information

**S1 Appendix. Delphi questionnaire items.**
(DOCX)

**S2 Appendix. Delphi expert panel.**
(DOCX)

**S1 File. Study dataset.**
(ZIP)

## Acknowledgments

The authors would like to acknowledge the Delphi panel experts of the ACOPREFERENCE study for their participation in the study (S2 Appendix).

We also thank Dr Cristina Vidal and Antonio Torres from Dynamic Science (Spain) for their editorial and medical writing support, funded by Boehringer Ingelheim, Spain. The authors would also like to thank Boehringer Ingelheim for sponsoring the study.

## Author Contributions

**Conceptualization:** Carlos Escobar, Xavier Borrás, Ramón Bover Freire, Carlos González-Juanatey, Miren Morillas, Alfonso Valle Muñoz, Juan José Gómez-Doblas.

**Investigation:** Carlos Escobar, Xavier Borrás, Ramón Bover Freire, Carlos González-Juanatey, Miren Morillas, Alfonso Valle Muñoz, Juan José Gómez-Doblas.

**Methodology:** Carlos Escobar, Xavier Borrás, Ramón Bover Freire, Carlos González-Juanatey, Miren Morillas, Alfonso Valle Muñoz, Juan José Gómez-Doblas.

**Visualization:** Carlos Escobar, Xavier Borrás, Ramón Bover Freire, Carlos González-Juanatey, Miren Morillas, Alfonso Valle Muñoz, Juan José Gómez-Doblas.

**Writing – original draft:** Carlos Escobar, Xavier Borrás, Ramón Bover Freire, Carlos González-Juanatey, Miren Morillas, Alfonso Valle Muñoz, Juan José Gómez-Doblas.

**Writing – review & editing:** Carlos Escobar, Xavier Borrás, Ramón Bover Freire, Carlos González-Juanatey, Miren Morillas, Alfonso Valle Muñoz, Juan José Gómez-Doblas.

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
