## [Decision Letter · Decision Letter 0]

8 Jan 2020

PONE-D-19-21146

A Delphi consensus on the management of oral anticoagulation in patients with non-valvular atrial fibrillation in Spain: ACOPREFERENCE study

PLOS ONE

Dear Dr. Escobar,

Thank you for submitting your manuscript to PLOS ONE. After careful consideration, we feel that it has merit but does not fully meet PLOS ONE’s publication criteria as it currently stands. Therefore, we invite you to submit a revised version of the manuscript that addresses the points raised during the review process.

The Delphi technique is used to systematically combine expert opinion in order to arrive at an informed group consensus on a complex problem.

Several studies in health education have used the traditional Delphi technique and the e Delpji technique to determine consensus in a number of important need areas, as for example clinical governance.

In principle, the Delphi is a group method that is administered by a researcher or research team that assembles a panel of experts, poses questions, synthesizes feedback and guides the group towards common ground. The Delphi is a method for organizing conflicting values and experiences and facilitates the incorporation of multiple opinions into consensus. (41,42) This is achieved using iterative rounds of sequential surveys interspersed with controlled feedback reports and the interpretation of experts' opinion.

The ACOPREFERENCE project is an interesting nationwide Spanish multicentre 2-round Delphi study to 93 seek expert opinion on the management of anticoagulation therapy for patients with NVAF.

The manuscript has important limitations in the results presentation:

1) the  panel of selected healthcare experts is crucial. In the Study Design (Delphi Process) and in the Results (Characteristic of panel expert) the authors should add:  %academic hospitals / number years of tenure in this academic hospital

2) in the Results and in the tables the different response rate in round 1 /2/3 are not clearly expressed (histograms could be very helpful).consequently the discussion is limited by these important two variables (selectio of experts and response rate)

I'd like to review the manuscript after these important revisions in the Results.

Minor revisions suggested by the two reviewers are required.

We would appreciate receiving your revised manuscript by Feb 20 2020 11:59PM. To enhance the reproducibility of your results, we recommend that if applicable you deposit your laboratory protocols in protocols.io, where a protocol can be assigned its own identifier (DOI) such that it can be cited independently in the future. For instructions see: http://journals.plos.org/plosone/s/submission-guidelines#loc-laboratory-protocols

We look forward to receiving your revised manuscript.

Kind regards,

Yan Li

Academic Editor

PLOS ONE

Journal Requirements:

2. Please clarify whether the panel members provided their written informed consent to participate in this study.

"I have read the journal's policy and the authors of this manuscript have the following competing interests: I have served on advisory boards for Boehringer, Pfizer/Bristol-Myers Squibb (BMS), Bayer and Daiichi Sankyo, and have also received speaker fees from these companies. C. González Juanatey has served on advisory boards for Boheringer, Astra-Zeneca, Novartis, and Bayer, and he has also received research funding from Abbott, Boheringer, Astra-Zeneca, and Pfizer. A. Valle Muñoz has served on advisory boards for MSD, Boheringher, Bayer, Daiichi, and Novartis and has received honoraria for providing expert testimony from MSD, Boheringher, Bayer, Daiichi, Novartis, Amgen, Sanofi, Pzifer, Janssen, Astrazeneca, Servier, and Rovi. J.J. Gómez Doblas has received honoraria for advisory services from Bayer, Astrazeneca, MSD, Daiichi, BMS, Amgen and Sanofi. The other authors declare that they have no competing interests. "

We note that you received funding from a commercial source: Boehringer Ingelheim

Reviewers' comments:

Reviewer's Responses to Questions

**Comments to the Author**

1. Is the manuscript technically sound, and do the data support the conclusions?

Reviewer #1: Yes

Reviewer #2: Yes

2. Has the statistical analysis been performed appropriately and rigorously? 

Reviewer #1: Yes

Reviewer #2: Yes

3. Have the authors made all data underlying the findings in their manuscript fully available?

Reviewer #1: Yes

Reviewer #2: Yes

4. Is the manuscript presented in an intelligible fashion and written in standard English?

Reviewer #1: Yes

Reviewer #2: Yes

5. Review Comments to the Author

Reviewer #1: Thank you for the invitation to review this manuscript, in this interesting paper, the authors described a Delphi consensus on the management of oral anticoagulation in patients with non-valvular atrial fibrillation in Spain, the article is in the well written and sound, however, some of the weakness bring to attention, please re-design or re-organize the tables, most of them are way too busy, description is too long and easy to lost, please try to use simple words or phrases to take place.

Reviewer #2: Just like the authors mentioned, the major concern is the bias from the selection of experts and the subjectivity linked to the potentially divergent personal opinions of the panellists which may partly result from unevenly distributed expertise. Overseas experts might help with the regional biased opinion.

6. PLOS authors have the option to publish the peer review history of their article (what does this mean?). If published, this will include your full peer review and any attached files.

Reviewer #1: No

Reviewer #2: No

---

## [Author Response · Author response to Decision Letter 0]

7 Mar 2020

We thank the academic editor and the two reviewers for their detailed and useful review of our manuscript. We appreciate the positive feedback, as well as the constructive suggestions to further improve our manuscript, and we have answered to each of their valuable comments. 

A rebuttal letter that responds to each point raised by the academic editor and reviewers is provided. This letter has been uploaded as a separate file and labeled "Response to reviewers".

---

## [Decision Letter · Decision Letter 1]

27 Mar 2020

A Delphi consensus on the management of oral anticoagulation in patients with non-valvular atrial fibrillation in Spain: ACOPREFERENCE study

PONE-D-19-21146R1

Dear Dr. Escobar

We are pleased to inform you that your manuscript has been judged scientifically suitable for publication and will be formally accepted for publication once it complies with all outstanding technical requirements.

With kind regards,

Maria Lucia Narducci, MD, PhD

Academic Editor

PLOS ONE

Additional Editor Comments (optional):

The revised version of the manuscript is now appropriately detailed.

Given the peculiar nature of a Delphi consensus, as indicated by a third reviewer, the study would benefit from disclosure of the potential conflict of interest of the panelists, in addition to that of the authors of the manuscript, in a separate Appendix.

Reviewers' comments:

Reviewer's Responses to Questions

**Comments to the Author**

1. If the authors have adequately addressed your comments raised in a previous round of review and you feel that this manuscript is now acceptable for publication, you may indicate that here to bypass the “Comments to the Author” section, enter your conflict of interest statement in the “Confidential to Editor” section, and submit your "Accept" recommendation.

Reviewer #3: (No Response)

2. Is the manuscript technically sound, and do the data support the conclusions?

Reviewer #3: Yes

3. Has the statistical analysis been performed appropriately and rigorously? 

Reviewer #3: Yes

4. Have the authors made all data underlying the findings in their manuscript fully available?

Reviewer #3: Yes

5. Is the manuscript presented in an intelligible fashion and written in standard English?

Reviewer #3: Yes

6. Review Comments to the Author

Reviewer #3: The present revised manuscript is aimed at evaluating the opinion of experts in the field of atrial fibrillation on several items regarding the management of anticoagulation in non-valvular atrial fibrillation.

Current guidelines on anticoagulation provide evidence-based recommendations for a limited number of relevant topics explicitly addressed in clinical trials. However, the complexity of real-world clinical practice often faces clinicians with uncertainties and difficult decisions. Based on these considerations, the authors identify the Delphi technique as a validated tool to develop expert opinions on controversial items and provide convincing evidence in support of their methodology.

The description of the process from controversial item selection, iterative consensus achievement, and final recommendations is appropriately detailed.

In conclusion, the authors of the present study provide several expert-based opinions on the management of oral anticoagulation in non-valvular atrial fibrillation, which may be of interest to the readers of Plos One.

However, given the peculiar nature of a Delphi consensus, it is my opinion that the study would benefit from disclosure of the potential conflict of interest of the panelists, in addition to that of the authors of the manuscript, in a separate Appendix.

7. PLOS authors have the option to publish the peer review history of their article (what does this mean?). If published, this will include your full peer review and any attached files.

Reviewer #3: No

---

## [Editor Report · Acceptance letter]

22 May 2020

PONE-D-19-21146R1 

A Delphi consensus on the management of oral anticoagulation in patients with non-valvular atrial fibrillation in Spain: ACOPREFERENCE study 

Dear Dr. Escobar:

I am pleased to inform you that your manuscript has been deemed suitable for publication in PLOS ONE. Congratulations! Your manuscript is now with our production department. 

With kind regards,

on behalf of

Dr. Maria Lucia Narducci 

Academic Editor

PLOS ONE